# Heterogeneous synaptic homeostasis: A novel mechanism boosting information propagation in the cortex

**Farhad Razi** [ID][1]*, **Belén Sancristóbal** [ID][2]

**1** Donders Institute for Brain, Cognition and Behaviour, Radboud University, Nijmegen, The Netherlands, **2** Computational Biology and Complex Systems Group, Department of Physics, Universitat Politècnica de Catalunya, Barcelona, Spain

* farhad.razi@donders.ru.nl

## Abstract

Perceptual awareness of auditory stimuli decreases from wakefulness to sleep, largely due to reduced cortical responsiveness. During wakefulness, neural responses to external stimuli in most cortical areas exhibit a broader spatiotemporal propagation pattern compared to deep sleep. A potential mechanism for this phenomenon is the synaptic upscaling of cortical excitatory connections during wakefulness, as posited by the synaptic homeostasis hypothesis. However, we argue that uniform synaptic upscaling alone cannot fully account for this observation. We propose a novel mechanism suggesting that the upscaling of excitatory connections between different cortical areas exceeds that within individual cortical areas during wakefulness. Our computational results demonstrate that the former promotes the transfer of neural responses and information, whereas the later has diminishing effects. These findings highlight the necessity of heterogeneous synaptic upscaling and suggest the presence of heterogeneity in receptor expression for neuromodulators involved in synaptic modulation along the dendrite.

**Data availability statement:** Code supporting the findings of this paper are publicly available

## Author summary

As we transition from wakefulness to sleep, our perception of the external world fades, and the brain's neural activity undergoes profound changes. Neurons not only alter their firing patterns, but the strength of their synaptic connections also weakens during sleep and increases upon waking. While this process, known as synaptic homeostasis, is gaining experimental validation, its causal link to the accompanying changes in brain activity and cognition are not fully elucidated. A hallmark of wakefulness is the ability of external stimuli to propagate across widespread cortical areas, engaging multiple sensory regions–something that is limited during sleep. Here, we use a computational model to show that uniformly increasing synaptic strength across different spatial scales does not

with permanent DOIs. Simulation routine code is available at
https://doi.org/10.5281/zenodo.15753856. iQuanta framework code is available at https://doi.org/10.5281/zenodo.15753882. All data generated during this study are included in the simulation routine repository (https://doi.org/10.5281/zenodo.15753856).

**Funding:** This research was funded by the Postdoctoral Junior Leader Fellowship Programme from La Caixa Banking Foundation (grant number LCF/BQ/PI18/11630004; https://fundacionlacaixa.org/) awarded to B.S. The funders had no role in study design, data collection and analysis, decision to publish, or preparation of the manuscript.

**Competing interests:** The authors have declared that no competing interests exist.

replicate this enhanced signal propagation. Instead, we find that selectively strengthening long-range excitatory connections–those linking distant regions–boosts signal spread more effectively than uniform changes, which primarily increase spontaneous activity and disrupts signal transmission. These findings refine the synaptic homeostasis hypothesis and highlight the role of spatial rules in shaping network connectivity to effectively modulate information processing across the sleep-wake cycle.

## Introduction

During the sleep-wake cycle (SWC), the capacity of the cerebral cortex to transmit neural signals across cortical areas–known as cortical effective connectivity [1]—is generally higher during wakefulness than during the deep phases of non-rapid eye movement (NREM) sleep [2–4]. While stimulation of the somatosensory cortex under anesthesia may not follow this general pattern–evoked slow oscillations can propagate broadly from frontal to occipital regions [5]–during NREM sleep, maximal cortical activation following sensorimotor stimulation remains local and only shifts spatially over time during wakefulness [6]. However, the precise neural mechanisms underlying this enhanced propagation of neural responses remain largely speculative.

Changes in neuronal and synaptic dynamics across the SWC within neural pathways could alter propagation patterns. Experimental evidence indicates that low concentrations of neuromodulators released from the ascending arousal network (AAN) during sleep [7] modulate neuronal dynamics [8,9] and synaptic strength [10–12]. The characteristic oscillating dynamics of neuronal transmembrane voltage, alternating between active (Up) and silent (Down) states during NREM sleep [8,9], might interrupt communication between cortical regions [6,13–15]. According to this view, evoked Down states following external perturbations in cortical neurons disrupt long-lasting causal interactions among cortical areas during NREM sleep. However, direct experimental evidence supporting this claim remains elusive, and certain empirical observations challenge it as the sole reason for altered cortical effective connectivity during sleep.

For instance, single and multi-unit recordings from the primary auditory cortex (A1) across various species [4,16–19] have revealed that evoked neural responses to auditory stimuli are comparable across the SWC, despite the aforementioned significant changes in cortical dynamics. It is only in higher-order cortical areas downstream from A1 where evoked neural responses are notably increased during wakefulness compared to NREM sleep [4,19]. This suggests that local changes in neuronal dynamics alone cannot fully account for the differences in response propagation in the cortex, indicating that variations in synaptic strength might also play a significant role.

Regarding changes in synaptic dynamics, the synaptic homeostasis hypothesis (SHY) [11,12] proposes that synaptic strength in many cortical circuits decreases during sleep to counterbalance the net synaptic upscaling observed during wakefulness. The increase in synaptic strength during wakefulness yields two opposing effects. Firstly, it amplifies stimulus-evoked postsynaptic currents due to the larger projecting axons of excitatory neurons compared to the more localized axons of inhibitory neurons [20–22]. Consequently, synaptic upscaling can enhance the amplitude of evoked responses in secondary sensory areas, facilitating the transmission of neural responses across the cortical hierarchy. Secondly, synaptic upscaling enhances spontaneous postsynaptic currents, which are not triggered by external stimuli. *In vitro* studies have shown that an increase in spontaneous synaptic currents, when balanced to avoid overexcitation or overinhibition, decreases the amplitude

of evoked responses to external stimuli [23]. Therefore, synaptic upscaling simultaneously enhances and diminishes the transmission of neural responses across different cortical areas, depending on the context. This phenomenon underscores the intricate balance of synaptic dynamics and its impact on neural response transmission and processing within the cerebral cortex.

We propose a mechanism that sets up a competition between these opposing effects on evoked neural activities: the *driving effect*, which enhances transmission by amplifying stimulus-evoked postsynaptic currents, and the *pulling effect*, which reduces transmission by increasing spontaneous postsynaptic currents. Uniform synaptic upscaling during wakefulness, without favoring the driving over the pulling effect, may not sufficiently explain the improved propagation of neural responses during wakefulness. The balance between these opposing effects is essential for understanding how response propagation and information processing occur within the cerebral cortex across different states of consciousness.

Hierarchical models of the cortex [24,25] distinguish between excitatory connections at the circuit level. Generally, inter-excitatory connections link different cortical areas, whereas intra-excitatory connections operate within individual cortical areas. Inter-excitatory connections typically entail a driving effect that facilitates downstream transmission of neural responses, whereas intra-excitatory connections involve modulatory synapses that control local neural activity and lead to a pulling effect.

In this paper, we introduce the *heterogeneous synaptic homeostasis hypothesis* at the circuit level, suggesting that synaptic upscaling should favor inter- over intra-excitatory connections. This approach allows the driving effect, which improves the transmission of neural responses across cortical areas, to prevail over the pulling effect caused by spontaneous postsynaptic currents. The concept of heterogeneous synaptic homeostasis provides a refined perspective on balancing the driving and pulling effects within cortical circuits, emphasizing the significance of the spatial organization of cortical networks that are state–sensitive and facilitate efficient information transmission.

To investigate this hypothesis, we employed a Wilson-Cowan model, which simulates the average firing rate of a cortical column [26]. The model replicates dynamics akin to those observed during NREM sleep and wakefulness [27]. It has been shown that synaptic upscaling of intra-excitatory connections, coupled with an increase in inhibitory synaptic strength that maintains the system's steady state near NREM sleep levels, leads to a gradual transition of the model's spontaneous activity from NREM-like to wakefulness-like dynamics [28]. Previous computational work modelling a thalamocortical network comprising multiple cortical columns [29] examined the emergent dynamics when excitatory-to-excitatory connection strength was modified and found that excessively strong excitation drives slow oscillations into a runaway regime, resulting in a sustained Up state. In this study, we examine how adjusting the synaptic upscaling of both intra- and inter-excitatory connections (by factors $\beta_{\text{intra}}$ and $\beta_{\text{inter}}$, respectively) influences evoked neural responses. Specifically, we study the responses of a single cortical column to stimuli with increasing intensity delivered via inter-excitatory connections. Finally, we explore a scenario where two cortical columns are symmetrically coupled by inter-excitatory connections. One column is perturbed while the other receives stimuli indirectly via inter-excitatory connections from the perturbed to the unperturbed column. We then analyze the effect of varying intra- and inter-synaptic upscaling on the propagation of neural responses between these columns.

Additionally, we establish a framework for quantifying stimulus-relevant information within evoked neural responses and investigate how intra- and inter-synaptic upscaling influence the amount of information that cortical populations convey about a stimulus and its propagation.

## Materials and methods

Our study adapts the Wilson–Cowan model [26] to simulate cortical column activity during both NREM sleep and wakefulness [27,28].

### One-cortical-column model

A cortical column is represented by interacting pyramidal ($p$) and inhibitory ($i$) neuronal populations (see Fig 1a). The temporal evolution of average membrane potentials ($V_{p/i}$) for each population was modeled using a conductance-based approach [30]:

$$\tau_p \dot{V}_p = -I_L^p - I_{\text{intra}}^p - I_{\text{GABA}}^p - I_{\text{KNa}}, \tag{1}$$

$$\tau_i \dot{V}_i = -I_L^i - I_{\text{intra}}^i - I_{\text{GABA}}^i. \tag{2}$$

Here, $\tau_k$ denotes the membrane time constant, while $I_L^k$, $I_{\text{intra}}^{p/i}$, and $I_{\text{inh}}^k$ represent the leak, intra-excitatory, and inhibitory currents, respectively, for population $k$ ($k \in \{p, i\}$) as follows:

$$I_L^k = \bar{g}_L (V_k - E_L^k), \tag{3}$$

$$I_{\text{intra}}^k = \beta_{\text{intra}} \, \bar{g}_{\text{AMPA}} s_{kp} (V_k - E_{\text{AMPA}}), \tag{4}$$

$$I_{\text{GABA}}^k = \beta_{\text{GABA}}^k \, \bar{g}_{\text{GABA}} s_{ki} (V_k - E_{\text{GABA}}). \tag{5}$$

For detailed parameter descriptions and values, refer to Tables 1 and 2. In brief, $\bar{g}_L$, $\bar{g}_{\text{AMPA}}$, and $\bar{g}_{\text{GABA}}$ represent average conductances for leak, AMPA-ergic, and GABA-ergic channels, respectively. $E_L^k$, $E_{\text{AMPA}}$, and $E_{\text{GABA}}$ denote their corresponding reversal potentials. $\beta_{\text{intra}}$ represents intra-excitatory synaptic upscaling, while $\beta_{\text{GABA}}^{p/i}$ adjusts inhibitory synaptic strength to counterbalance increased excitation due to the synaptic upscaling. This procedure is in line with inhibitory synaptic plasticity of neighboring excitatory synaptic plasticity [31,32]. $s_{kk'}$ represents the synaptic response in population $k$ due to presynaptic activity from population $k'$. It is formulated as the convolution of presynaptic firing rate $Q_{k'}$ with the average synaptic response, characterized by an alpha function with time constant $\gamma_{k'}$ [33], following the second-order differential equation:

$$\ddot{s}_{kk'} = \gamma_{k'}^2 \left( N_{kk'} \, Q_{k'}(V_{k'}) + \phi_k - s_{kk'} \right) - 2\gamma_{k'} \, \dot{s}_{kk'}. \tag{6}$$

$N_{kk'}$ represents the connectivity from population $k'$ to $k$. $\phi_k$ represents noise and is applied via intra-excitatory connections, simulated independently for each cortical population as a Gaussian process with zero autocorrelation time, zero mean, and 1.2 ms$^{-1}$ standard deviation. Firing rates of population $k$ is modeled using a sigmoid function of the average membrane potential [26] as:

$$Q_k(V_k) = \frac{Q_k^{\max}}{2} \left( 1 + \tanh\left( \frac{\pi}{2\sqrt{3}\,\sigma_k} (V_k - \theta_k) \right) \right). \tag{7}$$

Where $Q_k^{\max}$, $\theta_k$, and $\sigma_k$ represent the maximum firing rate, threshold, and inverse neural gain of population $k$, respectively.

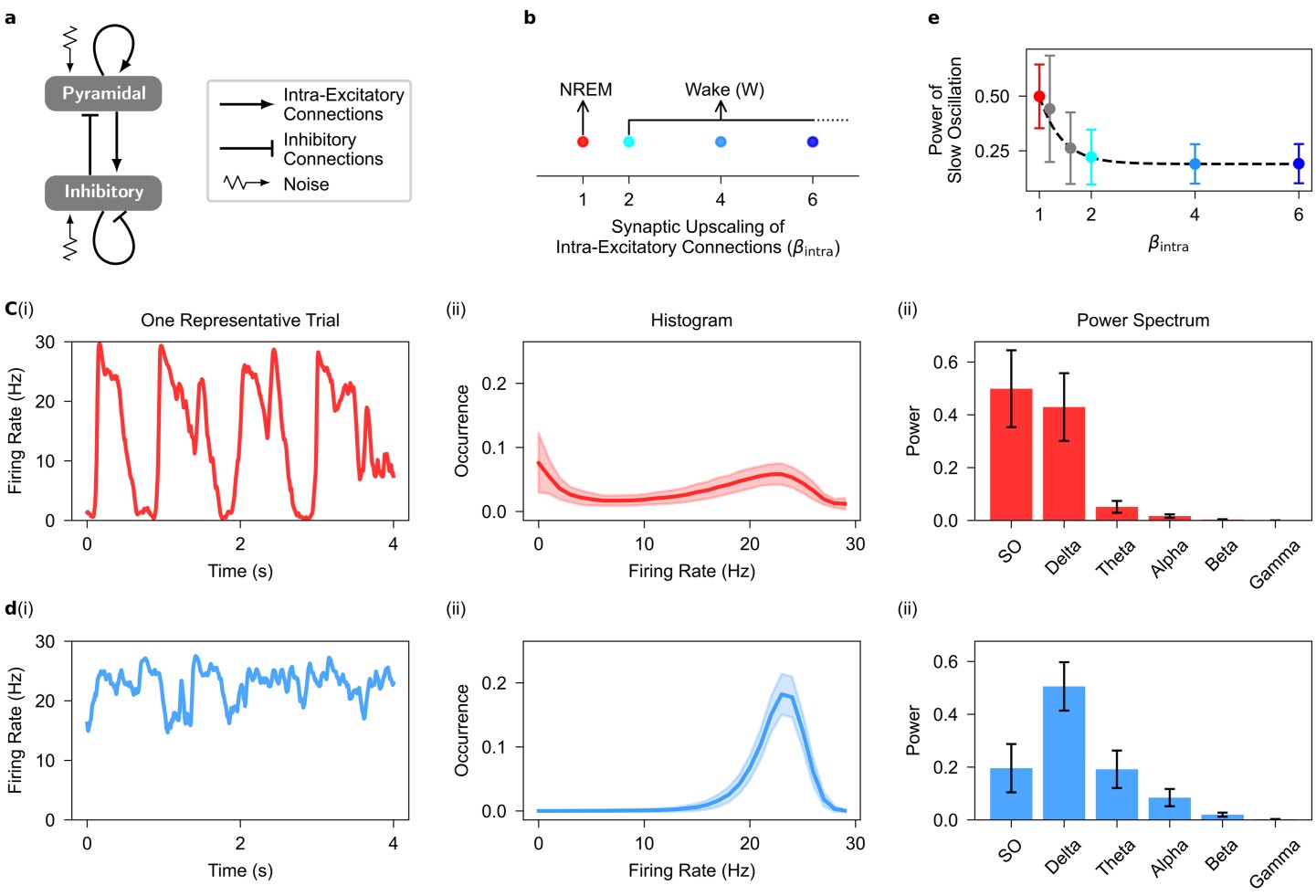

**Fig 1. Dynamical features of spontaneous firing activity in the one-cortical-column model. a**, Diagram of the one-cortical-column model containing one pyramidal and one inhibitory population, where each population receives independent noise. The couplings between pyramidal and inhibitory populations are intra-excitatory and inhibitory connections mediated through, respectively, intra-AMPAergic and GABAergic synapses (see Materials and methods). Refer Tables 1 and 2 for parameter description and values, respectively. **b**, Parameter space for synaptic upscaling of intra-excitatory connections ($\beta_{intra}$). **c**, Spontaneous firing rate signal for a representative trial (i), the distribution of firing rate signals (ii), and the power spectrum of signals (iii) when there is no intra-synaptic upscaling ($\beta_{intra} = 1$). The model produces electrophysiological features of NREM sleep when $\beta_{intra} = 1$. **d**, As in **c**, but for when intra-excitatory connections are upscaled ($\beta_{intra} = 2$). The model produces electrophysiological features of wakefulness when intra-excitatory connections are upscaled ($\beta_{intra} > 1$). **e**, The power ratio of slow oscillation (SO) gradually decreases with increasing intra-synaptic upscaling, color coded as in **b**. Shaded area and Error bar correspond to standard deviation over 500 trials. SO, <1 Hz; Delta, 1-4 Hz; Theta, 4-7 Hz; Alpha, 7-13 Hz; Beta, 3-30 Hz; Gamma, 30-100 Hz.

An activity-dependent potassium current, $I_{KNa}$, is included in pyramidal populations to produce NREM-like dynamics [34–36]:

$$I_{KNa} = \tau_p C_m^{-1} \bar{g}_{KNa} \frac{0.37}{1 + (\frac{38.7}{[Na]})^{3.5}} (V_p - E_K),\tag{8}$$

$$\tau_{Na} [\dot{Na}] = \alpha_{Na} Q_p(V_p) - Na_{pump}([Na]),\tag{9}$$

$$Na_{pump}([Na]) = R_{pump} \left( \frac{[Na]^3}{[Na]^3 + 3375} - \frac{[Na]_{eq}^3}{[Na]_{eq}^3 + 3375} \right).\tag{10}$$

**Table 1. Parameter description in the one-cortical-column model.**

| Symbol | Description |
|---|---|
| $Q_k^{\max}$ | Maximal firing rate of population $k$ |
| $\theta_k$ | Firing rate threshold of population $k$ (sigmoid function half activation) |
| $\sigma_k$ | Inverse neural gain of the sigmoid function of population $k$ |
| $\tau_k$ | Membrane time constant of population $k$ |
| $C_m$ | Membrane capacitance in the Hodgkin-Huxley model |
| $\phi_k$ | Gaussian noise on population $k$ |
| $N_{kk'}$ | Mean number of synaptic connections from population $k'$ to population $k$ |
| $\gamma_k$ | Time constant of the postsynaptic response of synapse type $k$ |
| $\bar{g}_X$ | Average X-ergic conductance |
| $E_X$ | Reversal potential of the X-ergic current |
| $\bar{g}_{KNa}$ | Average conductance of sodium-dependent potassium channel |
| $E_K$ | Nernst reversal potential of potassium channel |
| $\tau_{Na}$ | Time constant of sodium extrusion |
| $\alpha_{Na}$ | Sodium influx through population firing rate |
| $R_{pump}$ | Strength of the sodium pump |
| $Na_{eq}$ | Resting state sodium concentration equilibrium |
| $\beta_{intra}$ | Synaptic-upscaling factor for intra-excitatory connections |
| $\beta_{GABA}^k$ | Synaptic-upscaling factor for inhibitory connections on population $k$ |

The table is adapted from [27].

$\bar{g}_{KNa}$ and $E_K$ are the average conductance and reversal potential of the activity-dependent potassium channel. $C_m$ is membrane capacitance. $[Na]$ represents sodium concentration, with $\tau_{Na}$ as its extrusion time constant. $\alpha_{Na}$ denotes sodium influx due to firing, while $Na_{pump}([Na])$ represents sodium extrusion through pumps with strength $R_{pump}$. $[Na]_{eq}$ is the equilibrium sodium concentration.

The model exhibits NREM-like dynamics when $\beta_{intra} = 1$ (see Table 2 for parameter values). To simulate wakefulness, we increased intra-excitatory synaptic strength ($\beta_{intra} > 1$), aligning with SHY [11,12]. To prevent overexcitation, we increased inhibitory synaptic strength ($\beta_{GABA}^{p/i}$; see Table 3) in parallel with excitatory upscaling [31,32]. This adjustment keeps the steady-state average membrane potentials of both pyramidal ($V_p$) and inhibitory ($V_i$) populations constant, even as synaptic upscaling parameters change (see Computational pipeline). We set the steady-state membrane potential during wakefulness to match the Up state value observed in NREM sleep, as supported by electrophysiological evidence [9,37,38]. Notably, our findings remain robust when the steady-state potential is varied, as analyses using jittered values relative to the NREM Up state produce similar results (see S3 Fig).

Firing rates during NREM sleep are lower than in wakefulness [39] likely due to silent Down periods, as firing rates during Up states closely match those seen in wakefulness [40]. Our model captures this pattern (see Fig 1 and S2 Fig). While other experimental studies show that firing rates decrease as the UP state progresses [41], this feature is not explicitly examined in our current study.

A stability analysis of our model shows that both NREM-like and wake-like states correspond to stable fixed points. Their distinct dynamics arise from a more negative real eigenvalue in the wake-like state [28], which effectively suppresses oscillations in the presence of noise. This contrasts with models in which slow oscillations are generated by a limit cycle [42].

Stimuli are delivered via inter-excitatory connections, representing presynaptic firing of an unmodeled pyramidal population, $Q_p^{sti_{un}}$. In both NREM sleep and wakefulness, stimuli occur at random times relative to ongoing network activity, without targeting specific phases

**Table 2. Parameter values in the one-cortical-column model.**

| Symbol | Value | Unit |
|---|---|---|
| $Q_p^{\text{max}}$ | 30 | Hz |
| $Q_i^{\text{max}}$ | 60 | Hz |
| $\theta_p, \theta_i$ | −58.5 | mV |
| $\sigma_p$ | 6.7 | mV |
| $\sigma_i$ | 6 | mV |
| $\tau_p, \tau_i$ | 30 | ms |
| $C_m$ | 1 | $\mu\text{F/cm}^2$ |
| $\phi_{p/i}$ | 1.2 | $\text{ms}^{-1}$ |
| $N_{pp}$ | 144 | – |
| $N_{ip}$ | 36 | – |
| $N_{pi}$ | 160 | – |
| $N_{ii}$ | 40 | – |
| $\gamma_p$ | $70 \cdot 10^{-3}$ | $\text{ms}^{-1}$ |
| $\gamma_i$ | $58.6 \cdot 10^{-3}$ | $\text{ms}^{-1}$ |
| $\bar{g}_{\text{AMPA}}$ | 1 | ms |
| $\bar{g}_{\text{GABA}}$ | 1 | ms |
| $E_{\text{AMPA}}$ | 0 | mV |
| $E_{\text{GABA}}$ | −70 | mV |
| $E_L^p$ | −66 | mV |
| $E_L^i$ | −64 | mV |
| $\bar{g}_{\text{KNa}}$ | 1.9 | $\text{mS/cm}^2$ |
| $E_K$ | −100 | mV |
| $\tau_{\text{Na}}$ | 1.7 | ms |
| $\alpha_{\text{Na}}$ | 2 | $\text{mM} \cdot \text{ms}$ |
| $\bar{R}_{\text{pump}}$ | 0.09 | mM |
| $\text{Na}_{\text{eq}}$ | 9.5 | mM |
| $\beta_{\text{intra}}$ | 1 | – |
| $\beta_{\text{GABA}}^{p/i}$ | 1 | – |

The table is adapted from [27].

**Table 3. Parameter values of $\beta_{\text{GABA}}^k$ for intra-synaptic upscalings in wakefulness in the one-cortical-column model.**

| | $\beta_{\text{intra}} = 2$ | $\beta_{\text{intra}} = 4$ | $\beta_{\text{intra}} = 6$ |
|---|---|---|---|
| $\beta_{\text{GABA}}^p$ | 1.961 | 4.724 | 7.488 |
| $\beta_{\text{GABA}}^i$ | 2.165 | 4.65 | 7.134 |

such as Up states or Down states during NREM sleep. This external stimulation (see Fig 2a) induces an excitatory synaptic current $-I_{\text{inter}}^{p/i,\,\text{sti}}$ in both pyramidal and inhibitory populations that represents the stimulus-induced excitatory current and follows:

$$I_{\text{inter}}^{k,\,\text{sti}} = \beta_{\text{inter}}\,\bar{g}_{\text{AMPA}}s_{kp^{un}}\left(V_k - E_{\text{AMPA}}\right), \tag{11}$$

$$\ddot{s}_{kp^{un}} = \gamma_p^2\left(N_{kp^{un}}\,Q_{p^{un}}^{\text{sti}} - s_{kp^{un}}\right) - 2\gamma_p\,\dot{s}_{kp^{un}}. \tag{12}$$

$\beta_{\text{inter}}$ is the synaptic upscaling factor for the inter-excitatory connections. $N_{pp^{um}}$ and $N_{ip^{um}}$ represent the mean number of synaptic connections (16 and 4, respectively).

## Two-cortical-column model

We extended the model to two bidirectionally connected cortical columns (see Fig 4a), implementing symmetric excitatory connectivity between them (see Table 4), consistent with

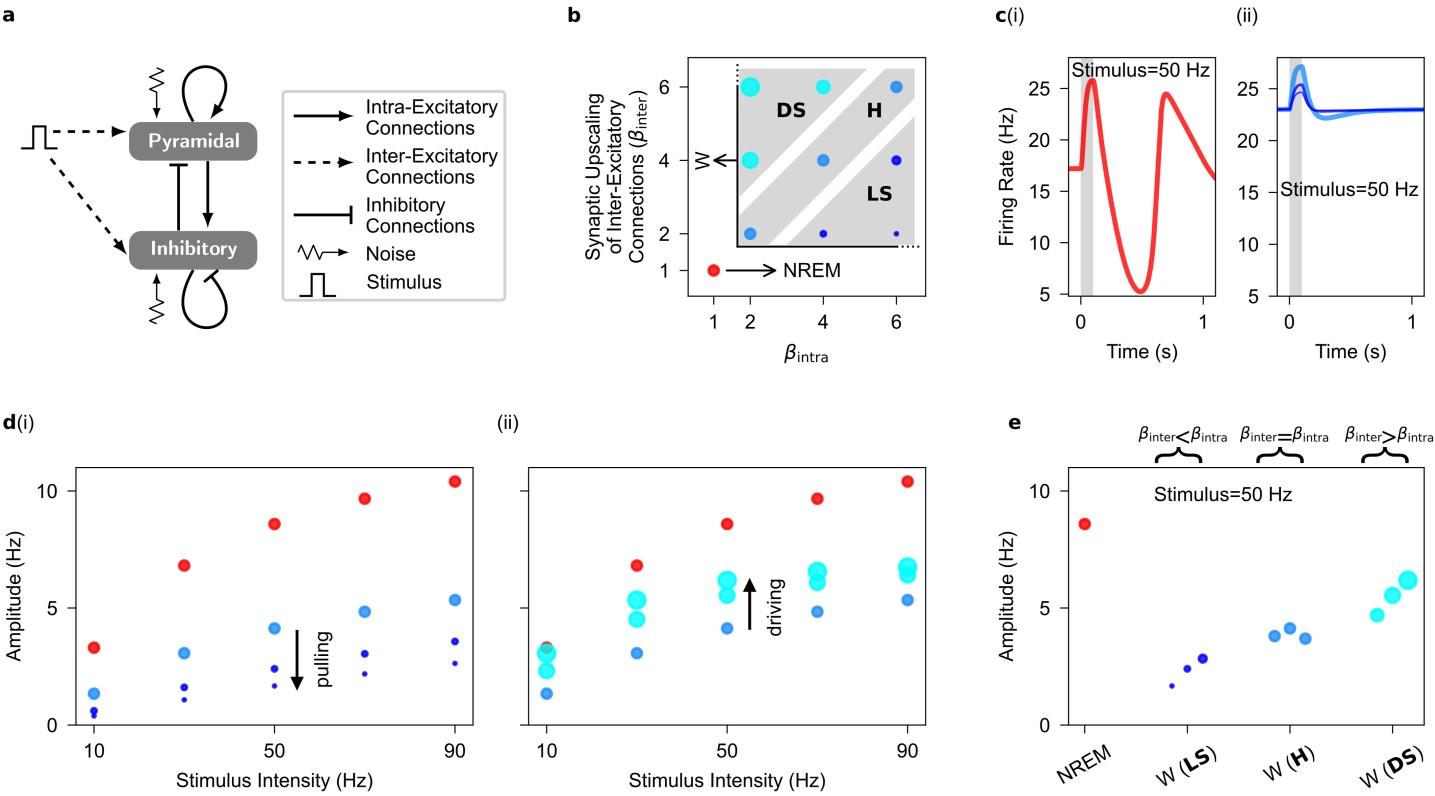

**Fig 2. Evoked firing responses to stimuli in the one-cortical-column model. a**, as in Fig 1**a**, but for when the model is subjected to stimuli. Stimuli are applied through inter-excitatory connections mediated through inter-AMPAergic synapses. Note that the noise term is set to zero for noise-free evoked response. **b**, as in Fig 1**b**, but for when intra- and inter-excitatory connections are upscaled in wakefulness. Note that synaptic upscaling in wakefulness can occur under three synaptic upscaling configurations: Local-Selective (LS: $\beta_{intra} > \beta_{inter} > 1$), Homogeneous (H: $\beta_{intra} = \beta_{inter} > 1$), and Distance-Selective (DS: $\beta_{inter} > \beta_{intra} > 1$). **c**, The evoked firing response in NREM sleep (i) and wakefulness (ii), color coded as in **b**, when the stimulus intensity is 50 Hz. The line width reflects the synaptic upscaling ratio, $\beta_{inter}/\beta_{intra}$. Shaded area corresponds to the stimulus duration. **d**, Increasing intra-synaptic upscaling (from $\beta_{intra} = 2$ to $\beta_{intra} = 6$) while inter-synaptic upscaling is constant ($\beta_{inter} = 2$) during wakefulness produces a pulling effect on the amplitude of evoked firing responses (i). On the other hand, increasing inter-synaptic upscaling (from $\beta_{inter} = 2$ to $\beta_{inter} = 6$) while intra-synaptic upscaling is constant ($\beta_{intra} = 2$) during wakefulness produces a driving effect on the amplitude of evoked firing responses (ii). **e**, The amplitude of evoked firing responses increases with increasing values of synaptic upscaling ratio, $\beta_{inter}/\beta_{intra}$, during wakefulness. Data points during wakefulness are organized based on increasing values of synaptic upscaling ratio on the x-axis. Note that in the homogeneous case, the value of $\beta_{inter}/\beta_{intra}$ is equal to one for all three data points. The amplitude of evoked firing responses increases as the synaptic upscaling transitions from local-selective (LS) to distance-selective (DS) upscaling during wakefulness.

previous studies [20,22,43–46]. The average membrane potentials in this expanded model evolve according to:

$$\tau_p \dot{V}_p = -I_L^p - I_{intra}^p - I_{GABA}^p - I_{KNa} - I_{inter}^p, \tag{13}$$

$$\tau_i \dot{V}_i = -I_L^i - I_{intra}^i - I_{GABA}^i - I_{inter}^i. \tag{14}$$

$I_{inter}^{p/i}$ represents the inter-column excitatory currents in our two-column model as:

$$I_{inter}^k = \beta_{inter}\, \bar{g}_{AMPA} s_{kp'} \left(V_k - E_{AMPA}\right), \tag{15}$$

$$\ddot{s}_{kp'} = \gamma_{p'}^2 \left(N_{kp'}\, Q_{p'}\left(V_{p'}\right) - s_{kp'}\right)\, - 2\gamma_{p'}\, \dot{s}_{kp'}. \tag{16}$$

**Table 4. Parameter values of connectivity in the two-cortical-column model.**

| Symbol | Value | Description |
| --- | --- | --- |
| $N_{pp'}$, $N_{p'p}$ | 16 | Mean number of synaptic connections from $p$ to $p'$ (and $p'$ to $p$) |
| $N_{ip'}$, $N_{i'p}$ | 4 | Mean number of synaptic connections from $p$ to $i'$ (and $p'$ to $i$) |

$p'$ represents the pyramidal population in the second cortical column. The second column's dynamics mirror the first, with population indices swapped ($p$ with $p'$ and $i$ with $i'$).

As before, synaptic upscaling maintains E/I balance between intra- and inter-synaptic upscalings (see Table 5). An external stimulus applied via inter-excitatory synapses (see Fig 4a) directly affects only one cortical column, referred to as the *perturbed column*, inducing an excitatory current $-I_{\mathrm{inter}}^{p/i,\,\mathrm{sti}}$, where $I_{\mathrm{inter}}^{p/i,\,\mathrm{sti}}$ follows Eq 11. The second column, referred to as the *unperturbed column*, receives the stimulus indirectly through inter-excitatory connections from the perturbed column.

## Computational pipeline

We implemented our simulations in Python, employing a stochastic Heun method [47] with a temporal resolution of 0.1 ms. The complete codebase is publicly accessible on GitHub [48]. Each trial was simulated independently, with all variables initialized using random values drawn from a uniform distribution. To ensure steady-state dynamics, we discarded the initial 4 seconds of each simulation to eliminate transients. Our analysis focused on the subsequent 4-second period of stabilized activity.

To obtain the value of $\beta_{\mathrm{GABA}}^{p/i}$ for each synaptic upscaling, we conducted 500 independent trials across the parameter space representing NREM sleep (see Table 2). From these simulations, we extracted the peak membrane potentials of pyramidal and inhibitory populations during Up states, identified as the active modes in the bimodal distribution of spontaneous firing activities. Down states, conversely, corresponded to the silent modes. We then calibrated the strength of inhibitory synapses (i.e., $\beta_{\mathrm{GABA}}^{p/i}$; see Table 3) for synaptic upscalings in wakefulness ($\beta_{\mathrm{intra}}$) to ensure that the steady-state average membrane potentials of both pyramidal ($V_p$) and inhibitory ($V_i$) populations matched their respective Up state values during NREM sleep. This approach maintains neuronal excitability across sleep-wake transitions and aligns with experimental observations that cortical neural activity during

**Table 5. Parameter values of $\beta_{\mathrm{GABA}}^k$, $k \in \{p,\, i,\, p',\, i'\}$, for various synaptic upscalings in the two-cortical-column model.**

| | $\beta_{\mathrm{intra}} = 1$ | $\beta_{\mathrm{intra}} = 2$ | $\beta_{\mathrm{intra}} = 4$ | $\beta_{\mathrm{intra}} = 6$ |
| --- | --- | --- | --- | --- |
| $\beta_{\mathrm{inter}} = 1$ | $\beta_{\mathrm{GABA}}^{p/p'} = 1.18$ $\beta_{\mathrm{GABA}}^{i/i'} = 1.149$ | n/a | n/a | n/a |
| $\beta_{\mathrm{inter}} = 2$ | n/a | $\beta_{\mathrm{GABA}}^{p/p'} = 2.268$ $\beta_{\mathrm{GABA}}^{i/i'} = 2.441$ | $\beta_{\mathrm{GABA}}^{p/p'} = 5.032$ $\beta_{\mathrm{GABA}}^{i/i'} = 4.926$ | $\beta_{\mathrm{GABA}}^{p/p'} = 7.795$ $\beta_{\mathrm{GABA}}^{i/i'} = 7.41$ |
| $\beta_{\mathrm{inter}} = 4$ | n/a | $\beta_{\mathrm{GABA}}^{p/p'} = 2.575$ $\beta_{\mathrm{GABA}}^{i/i'} = 2.717$ | $\beta_{\mathrm{GABA}}^{p/p'} = 5.339$ $\beta_{\mathrm{GABA}}^{i/i'} = 5.202$ | $\beta_{\mathrm{GABA}}^{p/p'} = 8.102$ $\beta_{\mathrm{GABA}}^{i/i'} = 7.686$ |
| $\beta_{\mathrm{inter}} = 6$ | n/a | $\beta_{\mathrm{GABA}}^{p/p'} = 2.882$ $\beta_{\mathrm{GABA}}^{i/i'} = 2.993$ | $\beta_{\mathrm{GABA}}^{p/p'} = 5.646$ $\beta_{\mathrm{GABA}}^{i/i'} = 5.478$ | $\beta_{\mathrm{GABA}}^{p/p'} = 8.409$ $\beta_{\mathrm{GABA}}^{i/i'} = 7.963$ |

Up states mirrors those during wakefulness [9,37,38]. This approach is also repeated for the two-cortical-column model (see Table 5).

Our study encompassed both stochastic and deterministic simulations. For stochastic simulations, we introduced Gaussian noise and performed 500 independent trials for each brain state, including NREM sleep and various synaptic upscalings ($\beta_{intra}$, $\beta_{inter}$ > 1) in wakefulness. Additionally, we conducted deterministic simulations without the Gaussian noise to evaluate the model's response amplitude to external stimuli under controlled conditions.

### Data analysis

All data analyses were conducted offline using Python. Our analysis primarily focused on the firing rate signals of the pyramidal populations.

**Analysis of spontaneous electrophysiological patterns.** The dynamic characteristics of spontaneous activities were assessed across 500 independent trials for each brain state (see Fig 1c, 1d). To quantify the amplitude variability of these spontaneous activities, we computed normalized distributions of firing rate signals for each distinct brain state (see Fig 1c(ii)). Additionally, we employed Welch's method [49] to generate spectrograms of the firing rate signals and to characterize the frequency content of spontaneous activities in each brain state (see Fig 1c(iii)).

**Analysis of evoked responses to stimuli.** The amplitude of evoked firing responses was extracted at the stimulus offset from the deterministic simulations by subtracting the prestimulus values (see Fig 2d). We also calculated synaptic excitation (E) and inhibition (I) on pyramidal populations across different brain states (see S4 Fig). It is important to note that in stochastic simulations, as the number of trials approaches infinity, the amplitudes of evoked responses and E/I values converge to those observed in deterministic simulations.

**Analysis of stimulus-related information.** We developed a comprehensive framework to assess stimulus-related information in stochastic evoked firing responses (see S1 Appendix). Our approach is grounded in the neuronal perspective of information as "a difference that makes a difference" [50]. To quantify this information in neural firing responses, we introduce two novel measures: *information detection* and *information differentiation*.

*Information detection* quantifies the statistical distinction between stimulus-evoked neural firing responses and spontaneous activities. This measure evaluates whether observed firing patterns can be reliably attributed to stimulus presentation. While crucial for perception, information detection alone does not ensure rich encoding of external stimuli in neural responses. *Information differentiation*, on the other hand, assesses the statistical dissimilarity among neural firing responses to various stimuli. This measure determines whether distinct firing patterns can be reliably associated with specific stimuli, indicating the neural system's capacity to discriminate between different stimuli.

High levels of information detection and differentiation facilitate precise decoding of stimulus features from neural firing patterns by an ideal observer possessing prior stimulus knowledge. To quantify the information content within stochastic evoked responses, we employ a diverse set of analytical approaches, including machine learning algorithms, statistical significance tests, and information-theoretic methods. For researchers interested in replicating or extending our analysis, we have made our custom Python module for information quantification, *iQuanta*, publicly available on GitHub [51].

**Unsupervised machine learning framework.** Our unsupervised machine learning framework utilized the K-means clustering algorithm.

Information detection in each brain state aims at distinguishing stochastic evoked firing responses to a specific stimulus intensity from spontaneous firing activities. To do so, data

consisted of 500 spontaneous firing rates sampled at 100 ms prior to stimulus onset and of 500 evoked firing rates obtained at stimulus offset for a given intensity. The number of clusters were set to two (S: spontaneous; E: evoked), with random initialization of centroids. Data were partitioned using stratified 10-fold cross-validation [52]. In each fold, cluster centroids were estimated from the training set and used to assign cluster labels to the test set. Clustering performance was quantified using Normalized Mutual Information (NMI) [53], calculated between predicted and true labels and averaged across folds. NMI quantifies the correspondence between predicted cluster labels and true labels, with values ranging from 0 (indicating random clustering) to 1 (denoting perfect clustering).

The true cluster labels are denoted as $C = \left\{ C_K^i \,\middle|\, i = 1, \dots, n; \ K \in \{S, E\} \right\}$, where $n = 100$ is the number of elements in the test set. The predicted cluster labels are denoted as $C' = \left\{ C_{K'}^i \,\middle|\, i = 1, \dots, n; \ K' \in \{S, E\} \right\}$. NMI measures the similarity between the predicted cluster labels, $C'$, and the true labels, $C$, by computing the mutual information, $\mathrm{MI}(C, C')$, as:

$$\mathrm{MI}(C, C') = \sum_{k \in \{S,E\}} \sum_{k' \in \{S,E\}} p(k, k') \log_2 \frac{p(k, k')}{p(k) p(k')} \tag{17}$$

Where $p(k)$ and $p(k')$ are, respectively, the probabilities that a randomly chosen data point in the test set is labeled within the true class $K \in \{S, E\}$, or is assigned the predicted label $K' \in \{S, E\}$. $p(k, k')$ is the joint probability that a randomly chosen data point in the test set has true label $K \in \{S, E\}$ and is predicted as $K' \in \{S, E\}$. $\mathrm{MI}(C, C')$ determines how much uncertainty is reduced about the true labels $C$ by knowing the predicted clusters $C'$ from the K-means clustering algorithm [54]. Finally, the NMI [53] is computed by normalizing the $\mathrm{MI}(C, C')$ as:

$$\mathrm{NMI}(C, C') = \frac{\mathrm{MI}(C, C')}{\sqrt{H(C) H(C')}}, \tag{18}$$

$$H(C) = - \sum_{k \in \{S,E\}} p(k) \log_2 p(k), \tag{19}$$

$$H(C') = - \sum_{k' \in \{S,E\}} p(k') \log_2 p(k') \tag{20}$$

Where $H(C)$ and $H(C')$ are, respectively, the entropies of the true $p(k)$ and predicted $p(k')$ label distributions. We report the average MNI across these 10 folds, accompanied by the 95% confidence interval, for information detection (see Fig 3a). Information differentiation in each brain state aims at distinguishing stochastic evoked firing responses to different stimulus intensities. The NMI is obtained as explained above but now data consisted of 500 evoked firing rates obtained at stimulus offset for each stimulus intensity and the number of clusters was set to the number of stimulus intensities ($N = 5$) (see Fig 3c).

For a comprehensive description of our supervised machine learning framework, significance tests, and information-theoretic approaches, please refer to Information Quantification.

## Results

We used a population rate model to simulate the activity of a single cortical column and its interaction with another symmetrically coupled column. By adjusting the strength of

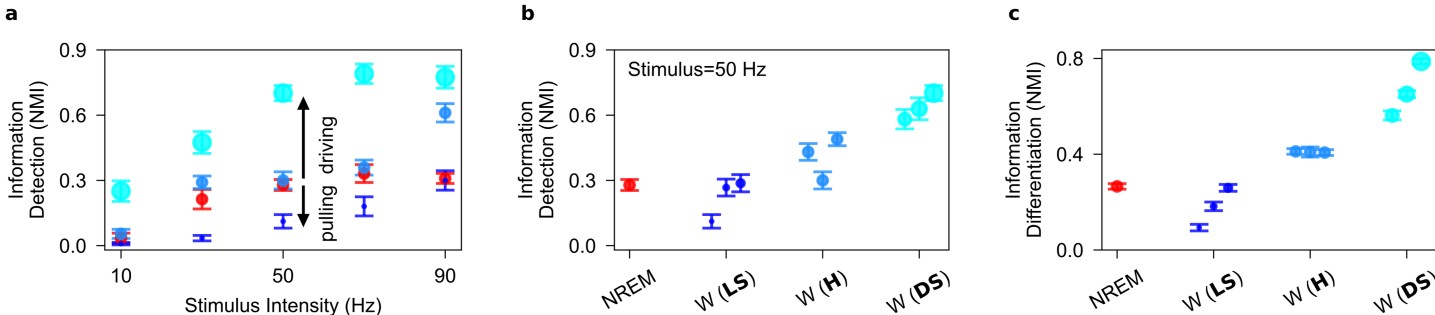

**Fig 3. Information content in the evoked firing responses to stimuli in the one-cortical-column model. a**, Increasing intra-synaptic upscaling while inter-synaptic upscaling is constant (from $\beta_{\text{intra}} = 2, \beta_{\text{inter}} = 2$ to $\beta_{\text{intra}} = 6, \beta_{\text{inter}} = 2$) during wakefulness produces a pulling effect on information detection. On the other hand, increasing inter-synaptic upscaling while intra-synaptic upscaling is constant (from $\beta_{\text{intra}} = 2, \beta_{\text{inter}} = 2$ to $\beta_{\text{intra}} = 2, \beta_{\text{inter}} = 6$) during wakefulness produces a driving effect on information detection. **b**, Information detection increases as the synaptic upscaling transitions from local-selective (LS) to distance-selective (DS) upscaling during wakefulness compared to NREM sleep. Note that data during wakefulness is organized based on increasing values of $\beta_{\text{inter}}/\beta_{\text{intra}}$ on the x-axis. **c**, As in **b**, but for information differentiation. Information differentiation increases as the synaptic upscaling transitions from local-selective (LS) to distance-selective (DS) upscaling during wakefulness compared to NREM sleep. Error bar corresponds to 95% confidence interval over 10 performance estimate of the K-means clustering algorithms.

excitatory synaptic coupling, we transitioned the model between NREM sleep and wakefulness states. Our analysis concentrated on the effects of synaptic upscaling during these states, particularly focusing on response amplitudes to transient stimuli and the encoding of stimulus intensity in firing responses of pyramidal populations.

## One-cortical-column model

A single cortical column is represented by a model comprising mutually coupled excitatory and inhibitory populations, each receiving independent Gaussian noise inputs (see Fig 1a).

**Electrophysiological patterns.** The model parameters (see Tables 1 and 2) were configured to generate spontaneous firing rates resembling NREM sleep. The parameter for intra-synaptic upscaling was set to 1 ($\beta_{\text{intra}} = 1$; see Fig 1b), reproducing neural dynamics akin to NREM sleep. These features include high-amplitude fluctuations (see Fig 1c(i)), a bimodal distribution (see Fig 1c(ii)), and high power in low-frequency bands (see Fig 1c(iii)) of the firing rate signals. These features remain robust even when the standard deviation of the noise in the model varies by up to 10% (see S1 fig).

By increasing the strength of intra-synaptic excitatory connections ($\beta_{\text{intra}} > 1$; see Fig 1b and Table 3), in line with SHY [11,12], and adjusting inhibitory strengths to prevent overexcitation (see Materials and methods and Table 3), our model gradually transitions from NREM sleep dynamics to wakefulness (see S2 fig and S3 fig). Firing rate signals become low amplitude (see Fig 1d(i)), show a unimodal distribution (see Fig 1d(ii)), and exhibit a relative increase in power at high-frequency bands (see Fig 1d(iii)). Additionally, the model shows a decrease in slow oscillation power (SO, 0.5–1 Hz), a key feature of NREM sleep, with increased intra-synaptic upscaling (see Fig 1e). Although our study focuses on two distinct patterns of electrophysiological activity, intermediate dynamics –where Down states become rare and are interspersed within sustained wake-like activity– are consitent with previous research [42]. As a consequence, power in the slow band gradually decreases from NREM-like to wake-like states, while delta power increases. These results demonstrate that the model effectively replicates well-established electrophysiological patterns observed during NREM sleep and wakefulness [8,9,34–36,39] as well as local sleep [42,55].

**Evoked responses to stimuli.** To analyze evoked responses, we applied transient stimuli to the cortical column through inter-excitatory connections (see Fig 2a). These stimuli represent presynaptic firing from an unmodeled upstream pyramidal population and vary in frequency from 10 Hz to 90 Hz in 20 Hz steps (see Materials and methods). Synaptic upscaling was implemented using two distinct scaling factors: $\beta_{intra}$, which modulates intra-synaptic connections within the modeled column, and $\beta_{inter}$, which affects the inter-synaptic connections from the unmodeled upstream pyramidal population to the modeled column. We examined various combinations of $\beta_{intra}$ and $\beta_{inter}$ (see Fig 2b) to assess their impact on evoked responses in the modeled cortical column.

In the absence of noise, distinct evoked responses are observed during NREM sleep and wakefulness, consistent with prior experimental observations [17]. During NREM sleep, where synaptic upscaling is absent ($\beta_{intra} = \beta_{inter} = 1$), responses exhibit a wave pattern characterized by an initial surge in firing rate followed by a subsequent decrease below the prestimulus equilibrium, maintaining a steady state well after stimulus offset (see Fig 2c(i)). In contrast, during wakefulness, synaptic upscaling ($\beta_{intra}$, $\beta_{inter} > 1$) substantially reduces the suppression of neuronal firing following activation (see Fig 2c(ii)).

For all combinations of $\beta_{intra}$ and $\beta_{inter}$, we quantified response amplitudes at stimulus offset. As seen in Fig 2d, response amplitudes increase with stimulus intensity during both NREM sleep ($\beta_{intra} = \beta_{inter} = 1$) and wakefulness ($\beta_{intra}$, $\beta_{inter} > 1$), with NREM sleep generally producing larger amplitudes (red dots in Fig 2d). Moreover, during wakefulness, keeping $\beta_{inter}$ constant and increasing $\beta_{intra}$ decreases evoked response amplitudes due to the aforementioned pulling effect (e.g., $\beta_{inter} = 2$ and $\beta_{intra}$ from 2 to 6, shown by progressively smaller dots in Fig 2d(i)). This pulling effect for intra-synaptic upscaling aligns with the reduced amplitude of evoked responses observed in *in vitro* studies as spontaneous postsynaptic currents increase [23].

Conversely, keeping $\beta_{intra}$ constant and increasing $\beta_{inter}$ increases evoked response amplitudes due to the driving effect (e.g., $\beta_{intra} = 2$ and $\beta_{inter}$ from 2 to 6, shown by progressively larger dots in Fig 2d(ii)). Notably, in the single cortical column architecture, increasing $\beta_{inter}$ exclusively modulates synapses conveying external stimuli, thus not affecting spontaneous postsynaptic currents and preventing a pulling effect.

While both intra- and inter-synaptic upscalings enhance excitatory synaptic currents upon stimulation, the net evoked synaptic current, quantified as $|E| - |I|$, decreases with increasing $\beta_{intra}$ and increases with increasing $\beta_{inter}$ (see panel a in S4 fig). This highlights the distinct effects of intra- and inter-synaptic upscaling on the evoked synaptic currents during the sleep-waking transition (see panel b in S4 fig). Modulating $\beta_{intra}$ and $\beta_{inter}$ independently allows us to define three distinct synaptic upscaling configurations characterizing the NREM-to-wakefulness transition (see Fig 2b). These are distinguished by the synaptic upscaling ratio, $\beta_{inter}/\beta_{intra}$, as follows:

1. Local-selective upscaling (LS): characterized by $\beta_{inter}/\beta_{intra} < 1$.
2. Homogeneous upscaling (H): characterized by $\beta_{inter}/\beta_{intra} = 1$.
3. Distance-selective upscaling (DS): characterized by $\beta_{inter}/\beta_{intra} > 1$.

Presenting findings based on $\beta_{inter}/\beta_{intra}$ provides a clearer representation than individual parameters. Firing rate response amplitudes during wakefulness increase with higher synaptic upscaling ratios (see Fig 2e and S5 fig), reaching maximum values for the DS policy.

**Stimulus-related information.** We quantify stimulus-related information in population firing rates by comparing information detection and differentiation (see Materials and methods and S1 Appendix). Information detection quantifies the performance of an optimal

classifier in distinguishing evoked responses from spontaneous firing activities. Information differentiation quantifies the performance of an optimal classifier in distinguishing evoked responses elicited by different stimulus intensities from one another. Both metrics are calculated using Normalized Mutual Information (NMI), which reflects classification performance on a scale from 0 (chance level) to 1 (perfect separation) (see Materials and methods).

Information detection increases with stimulus intensity during both NREM sleep and wakefulness (see Fig 3a). While, increasing $\beta_{intra}$ decreases information detection (from blue to dark blue dots in Fig 3a), increasing $\beta_{inter}$ increases information detection (from blue to light blue dots in Fig 3a). Our results show that it is DS policy during wakefulness that enhances information detection beyond NREM levels (see Fig 3b and see S6 fig).

Importantly, the higher trial-averaged amplitudes of evoked responses during NREM sleep compared to wakefulness need to be examined with increased variability across trials. Indeed, the distribution of evoked responses and spontaneous firing activities remain less distinguishable during NREM sleep than wakefulness, compromising information detection in NREM sleep.

Information differentiation during wakefulness exceeds NREM sleep levels under the DS policy (see Fig 3c). It decreases with increased intra-synaptic upscaling, but improves with inter-synaptic upscaling (as indicated by the larger dots in Fig 3c). This shows consistent encoding of stimulus intensity in firing rates during wakefulness compared to NREM sleep, underscoring the significance of heterogeneous synaptic upscaling favoring inter- over intra-synaptic connections. Additionally, these results are consistent with recent studies indicating that desynchronized cortical activity, as seen during wakefulness, enhances both the responsiveness and selectivity to tone's frequency compared to the synchronized activity typical of NREM-like states [56]. This underscores the critical role of cortical state in modulating information encoding.

## Two-cortical-column model

In this section, we investigate the dynamics of an extended model consisting of two identical cortical columns symmetrically coupled by inter-excitatory connections (see Fig 4a and Table 4).

**Electrophysiological patterns.** The analysis of spontaneous firing activities in the two-cortical-column model replicates the findings of the single-cortical-column model. Increasing $\beta_{intra}$ and $\beta_{inter}$ to mimic synaptic upscaling from sleep to wakefulness (see Table 5) reduces the amplitude of spontaneous fluctuations in pyramidal neuron firing rates (see S7 fig). Notably, the power of the SO band during NREM sleep decreases compared to the single cortical column scenario (see panel a in S7 fig and Fig 1c), supporting experimental findings that cortical de-afferentiation enhances NREM-like dynamics [14].

**Evoked responses to stimuli.** The two-cortical-column architecture serves as a model for exploring downstream information processing from a primary to a secondary cortical sensory area. One column, termed the *perturbed* cortical column, directly receives a stimulus, akin to a primary sensory area receiving direct unimodal thalamic signals. The stimuli modeled in the one-cortical-column model are used here as well. The other column, termed the *unperturbed* cortical column, detects the stimulus solely through presynaptic connections from the perturbed cortical column, mirroring higher-order unimodal sensory areas.

In the absence of noise, the evoked responses of both perturbed and unperturbed populations at stimulus offset reproduce those observed in the single-column model (see S8 fig and Fig 2d, 2e). Increasing $\beta_{intra}$ reduces response amplitudes in both populations, while increasing $\beta_{inter}$ enhances them (as indicated by the larger dots in panel a in S8 fig). Transitioning

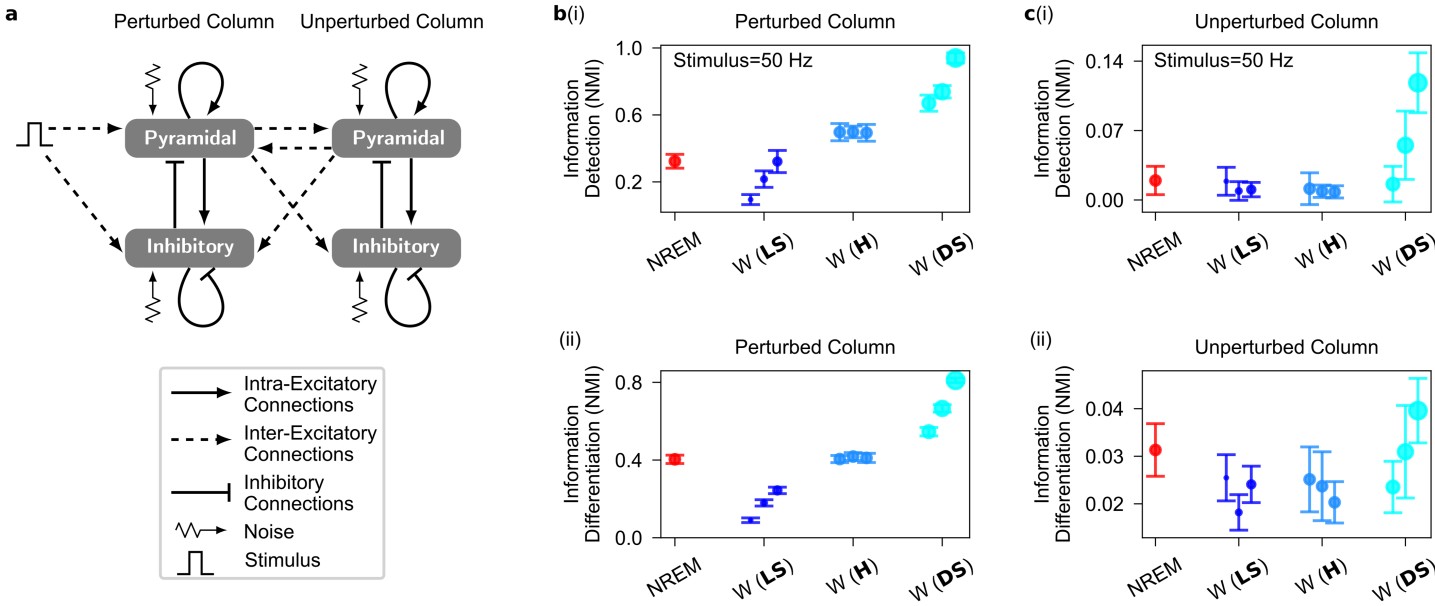

**Fig 4. Evoked firing responses to stimuli in the two-cortical-column model. a**, Diagram of the two-cortical-column model, where each population receives independent noise. The couplings between the two columns are symmetric and are inter-excitatory connections mediated through inter-AMPAergic synapses (see Materials and methods and Tables 1, 4 and 5 for symbol description and parameter values). In the context of spontaneous firing activity, stimulus intensity is set to zero. Note that the noise term is set to zero for noise-free evoked response. **b**, Information detection (i) and differentiation (ii) in the perturbed cortical column increase as the synaptic upscaling transitions from local-selective to distance-selective upscaling during wakefulness compared to NREM sleep. **c**, As in **b**, but for the unperturbed cortical column. Information detection (i) and differentiation (ii) in the unperturbed cortical column increase as the synaptic upscaling transitions from local-selective to distance-selective upscaling during wakefulness compared to NREM sleep. Error bar corresponds to 95% confidence interval over 10 performance estimate of the K-means clustering algorithms.

from LS to DS upscaling, by increasing the $\beta_{inter}/\beta_{intra}$ ratio, boosts response amplitude to external stimuli in both populations (see panel b in S8 fig).

In the two-cortical-column model, both intra- and inter-synaptic upscaling boost spontaneous synaptic activity in the unperturbed population, indicating a *pulling* effect now exerted by inter-excitatory connections. Nonetheless, inter-synaptic upscaling generates an overall driving effect as the net evoked synaptic current increases with increasing inter-synaptic upscaling, contrasting with intra-synaptic upscaling (see panel c in S8 fig).

**Stimulus-related information.** Encoding of stimulus intensity in the firing rate of the perturbed population increases from LS to DS upscaling (see Fig 4b and panel a in S9 fig). DS upscaling is the only policy that increases information beyond NREM sleep levels. In the unperturbed population, both information detection and differentiation greatly surpass NREM sleep levels during DS upscaling (see Fig 4c and panel b in S9 fig).

These results highlight the necessity for synaptic upscaling across the SWC to be spatially heterogeneous. Specifically, synapses between distinct cortical areas (inter-synapses) must be upscaled more than recurrent synapses within cortical areas (intra-synapses) throughout the SWC. This heterogeneity ensures better stimulus-encoded information during wakefulness compared to NREM sleep across the sensory processing chain.

## Robustness of the computational results

To quantify information content, we employed unsupervised machine learning techniques, such as K-means clustering algorithms. Our results remain consistent applying

supervised machine learning techniques, such as logistic classification algorithms (see panel a in S10 fig).

Furthermore, our findings are robust across different analytical approaches. Significance tests qualitatively reproduce findings on information detection and differentiation (see S1 Appendix and panel b in S10 fig). Moreover, using information theory to compute the mutual information (MI) between the distribution of evoked responses at the stimulus offset and the distribution of stimuli reveals that MI increases as synaptic upscaling transitions from LS to DS upscaling policy (see panel c in S10 fig).

## Discussion

Substantial differences exist in the discharge pattern of the AAN in the brainstem across various states of consciousness [57,58], resulting in alterations in neuromodulator concentrations throughout the brain [7]. These molecular changes impact neuronal [8,9] and synaptic [10–12] dynamics, potentially altering the cortex's ability to efficiently transmit neural signals [1–4]. Nevertheless, the direct causal relationships between these biological layers remain unclear.

This research investigates the relationship between the synaptic upscaling of excitatory connections [10–12] and enhanced cortical effective connectivity [2–4] during the transition from NREM sleep to wakefulness. Through computational modeling, we offer insights into how synaptic upscaling of excitatory connections not only induces dynamic changes in the electrical activity of the neural networks but also alters information propagation across these networks. Our results show that a spatially broader propagation of information and neural responses occurs during wakefulness compared to NREM sleep, provided that synaptic upscaling between distinct networks surpasses that of local and recurrent connections.

Our result aligns with several previously published computational studies. Firstly, the strengthening of inter-areal excitatory connections has been shown to enhance signal transmission in a network model of the macaque cortex [59]. Secondly, this outcome aligns with findings that rare long-range connections are necessary for information processing [60].

Our study concentrates on the interaction between two cortical columns, excluding whole-brain interactions. A more detailed model including various cortical and subcortical structures might offer additional insights into the spatial distribution of synaptic scaling. In our simplified model, the connections between the two cortical columns are excitatory and symmetrical, potentially not capturing the full complexity of structural connectivity across all cortical regions. Future work could benefit from distinguishing between feedforward and feedback excitatory connections to better understand how distinct modulations of synaptic upscaling affect propagation of information and neural response. Nonetheless, these would not invalidate the core concept of the DS synaptic upscaling policy as our conclusions hold true even when limited to a single cortical column, showing that DS synaptic upscaling improves stimulus-induced information encoding. Moreover, our neural mass model presupposes that neural communication is based on rate coding. Future investigations could explore spiking-based models since our hypothesis is not conditioned by the coding scheme (temporal or rate based).

A significant contribution of our study is the development of a framework for measuring information content within neural signals. Most studies in sleep research have not explicitly evaluated information content and are primarily based on the amplitude, latency and spectral characteristics of event related potentials. These metrics are used to infer whether the cortex detects stimuli [61,62], but not how much information regarding stimulus features is being encoded. Our work addresses this gap by showing that neither information encoding nor

its propagation are enhanced in wakefulness over NREM sleep, except when inter-synaptic upscalings surpass intra-synaptic upscalings. The robustness of our findings is supported by employing a variety of analytical methods, including unsupervised and supervised machine learning techniques, alongside statistical tests and information theory.

Recent studies highlight the significance of analyzing informational content. Research using Neuropixels probes in mice shows that burst firing in thalamic neurons and amplitude of cortical responses to electrical stimulation are highest during quiet wakefulness and lowest during anesthesia [63]. Since thalamic neurons tend to fire in bursts during NREM sleep [64–66], we might expect similar high activity levels in thalamic relay neurons during NREM sleep as during quiet wakefulness, despite reduced cortical response amplitudes. Our research suggests that distinguishing between information detection and differentiation in stimulus-evoked activity could explain how cortico-thalamo-cortical connections adjust information transfer during quiet wakefulness and NREM sleep.

Wakefulness is associated with consciousness and the capacity to respond to environmental stimuli, whereas sleep diminishes sensory perception [67–69]. Human EEG studies show the sleeping brain can perform basic auditory tasks, although higher cognitive functions are compromised. For instance, cognitive response to the subjects' own name during sleep is similar to that observed during wakefulness [70], whereas motor preparation in response to auditory stimuli are attenuated during NREM sleep relative to wakefulness [68]. Moreover, sensory encoding of intelligible stories attenuates during NREM sleep compared to wakefulness, despite comparable encoding of unintelligible stories [69]. These results point toward the diminished capacity of the brain to process sensory information with a higher cognitive demand during NREM sleep. Our findings suggest that heterogeneous synaptic upscaling from sleep to wakefulness enhances information detection and differentiation across a broader region of the cortex, allowing for more complex cognitive computations. Moreover, the gradual transition from a predominant Up state with rare Down states to alternating Up and Down states mimics local sleep and provides a framework for extending our model of spatially dependent homeostatic regulation to incorporate temporal upscaling dynamics and simulate the buildup of sleep pressure [55].

Our findings suggest reevaluating the SHY with a focus on circuit-level heterogeneity. Although our results need further empirical support, evidence exists for cellular-level heterogeneity in synaptic upscaling, particularly between perforated and non-perforated synapses. Perforated synapses are larger with discontinuous post-synaptic densities (PSDs), while non-perforated ones are smaller with continuous PSDs [71]. Perforated synapses in the mouse cerebral cortex expand their axon-spine contact area upon waking, unlike non-perforated synapses [71]. This structural difference highlights a selective approach to synaptic homeostasis at the cellular level. Moreover, sleep has been shown to modulate connectivity among neurons in local cortical networks of mice [72]. Although excitatory and inhibitory connections did not differ dramatically across natural SWC, increases in connectivity strength primarily occurred during prolonged wakefulness rather than NREM sleep. This effect reflects the strong homeostatic regulation of sleep following deprivation. Further experiments are needed to determine whether heterogeneity in synaptic homeostasis at the cellular-level extends to circuit-level connectivity.

Heterogeneous synaptic upscaling increases the recruitment of a wider network of neural populations across the cortical hierarchy during wakefulness compared to sleep, which is necessary for the emergence of various collective computations within networks of interconnected neurons [73]. Yet, the mechanism behind heterogeneous synaptic homeostasis remains unclear. We offer a speculative explanation: intra-synaptic and inter-synaptic

connections lie on different dendritic segments, each with a distinct receptor density for neuromodulators secreted by the AAN. Thus, the heterogeneity in receptor expressions results in the heterogeneous synaptic homeostasis during the SWC.

In a wider perspective, heterogeneity seems to be the norm rather than the exception within the brain. For instance, neural firing in pyramidal cells differs based on their target destinations, indicating heterogeneity within traditional pyramidal cell types [74]. Moreover, the developmental and regional distribution of N-methyl-D-aspartate (NMDA) receptors have been observed to be heterogeneous [75].

Research indicates that neural heterogeneity plays a functional role in the brain. It improves information transfer in spiking neural networks [76], enhances coding efficiency in predictive coding models [77], acts as a homeostatic mechanism preventing seizures [78], and supports stable learning in recurrent neural networks [79]. Our findings contribute to this field by showing that the increased cortical effective connectivity observed during wakefulness relative to NREM sleep may arise from heterogeneous synaptic homeostasis.

## Conclusion

Our study advances the understanding of how the spatial organization of synaptic strength shapes cortical information processing across the SWC. Using a combination of computational modeling, information-theoretic analysis, and machine learning tools, we show that the enhanced propagation of stimulus-related information observed during wakefulness cannot be explained by uniform synaptic upscaling alone, as proposed by the synaptic homeostasis hypothesis. Instead, we demonstrate that selectively strengthening excitatory connections between cortical areas –rather than within local circuits– enables more widespread and efficient information transmission.

This reveals a dual effect of synaptic upscaling: while local increases in synaptic strength raise spontaneous firing rates and impair signal propagation, selectively upscaling long-range connections promotes both the transfer and differentiation of stimulus-driven activity. These findings challenge conventional interpretations of the synaptic homeostasis hypothesis and highlight the critical role of spatial heterogeneity in synaptic changes. Our results suggest that *where* synaptic changes occur is as important as *how much* they change.

Furthermore, we introduce a comprehensive computational and analytical framework to quantify information encoding in neural signals –addressing a key limitation in sleep research, which has traditionally focused on signal amplitude rather than information content. Our model reproduces both classic electrophysiological markers of the SWC and recent observations on the spatial extent of cortical responses to stimuli. By assessing information detection and differentiation, we propose measures that more accurately reflect cortical communication and thus, cognitive function, than traditional trial-averaged metrics.

Future work could expand on this framework by exploring whole-brain models that incorporate dynamic changes in synaptic upscaling and parameter heterogeneity. Importantly, experimental validation of our distance selective upscaling mechanism remains a key avenue for future investigation.

## Supporting information

**S1 Fig. Dynamical features of spontaneous firing activity in the one-cortical-column model are robust to the changes in the standard deviation of the noise, $\phi$. a**, Spontaneous firing rate signal for a representative trial (i), the distribution of firing rate signals (ii), and the power spectrum of signals (iii) when $\phi = 0.9$ ms$^{-1}$. **b**, **c**, and **d**, As in **a**, but for when $\phi$ increases. Panel **c** here is as Fig 1**c**. Note that $\phi = 1.2$ ms$^{-1}$ is used as the value of the standard

deviation of the noise in this computational study. Shaded area and Error bar correspond to standard deviation over 500 trials.
(TIF)

**S2 Fig. Dynamical features of spontaneous firing activity in the one-cortical-column model changes with increasing intra-synaptic upscaling, $\beta_{\textbf{intra}}$. a**, Spontaneous firing rate signal for a representative trial (i), the distribution of firing rate signals (ii), and the power spectrum of signals (iii) when there is no intra-synaptic upscaling ($\beta_{intra} = 1$). Panel **a** here is as Fig 1**c**. **b, c, d, e**, and **f**, As in **a**, but for when intra-synaptic upscaling ($\beta_{intra}$) increases. Panel **d** here is as Fig 1**d**. Shaded area and Error bar correspond to standard deviation over 500 trials.
(TIF)

**S3 Fig. Robustness of evoked firing responses to variations in steady-state membrane potential. a**, Spontaneous firing rate signal for the representative trial shown in Fig 1**c(i)**. **b**, Same as in **a**, but with intra-excitatory connections upscaled ($\beta_{intra} = 2$). The steady-state membrane potential is set either higher (i) or lower (ii) than the Up state value during NREM sleep (red dashed horizontal line) by adjusting inhibitory synaptic strength $\beta_{GABA}^{k}$ below and above the values shown in Table 5, respectively. **c**, Amplitude of evoked firing responses as a function of the synaptic upscaling ratio, $\beta_{inter}/\beta_{intra}$, during wakefulness. Note that regardless of whether the steady-state membrane potential is fixed at values either higher (i) or lower (ii) than the Up state during NREM sleep, we reproduce the behavior shown in Fig 2**e**.
(TIF)

**S4 Fig. Effect of intra- and inter-synaptic upscaling on the net evoked synaptic currents in the one-cortical-column model. a**, The net evoked synaptic current (i), quantified as $|E| - |I|$, decreases with increasing $\beta_{intra}$ as opposed to when $\beta_{inter}$ increases in wakefulness. The line width reflects the synaptic upscaling ratio, $\beta_{inter}/\beta_{intra}$. Changes in the time trace of net evoked synaptic currrent determine changes in the time trace of evoked firing responses (ii). Note that the net synaptic current remains constant before stimulus onset across various synaptic upscaling scenarios, illustrating that synaptic upscaling is implemented in a configuration without causing predominant excitation or inhibition. Shaded area corresponds to the stimulus duration. **b**, Effects of $\beta_{intra}$ and $\beta_{inter}$ on the net evoked synaptic currents explain the pulling and driving effects associated with the intra- and inter-synaptic upscalings in wakefulness. Intra-synaptic upscaling decreases the net evoked synaptic current (i) that results in the decreased evoked responses (ii). Conversely, inter-synaptic upscaling increases the net evoked synaptic current (i) that results in the increased evoked responses (ii). Changes in the net evoked synaptic currrent determine changes in the amplitude of evoked firing responses (iii). Note that analysis in **b**(i) are carried out on the data points at stimulus offset.
(TIF)

**S5 Fig. The amplitude of evoked firing responses in the one-cortical-column model. a**, The amplitude of evoked firing responses increases with increasing values of synaptic upscaling ratio, $\beta_{inter}/\beta_{intra}$, during wakefulness when the stimulus intensity is 10 Hz (**a**), 30 Hz (**b**), 70 Hz (**c**) and 90 Hz (**d**). Note that the overall enhancement of the amplitude of evoked responses as the stimulus intensity increases from **a** to **d**.
(TIF)

**S6 Fig. Information detection in the one-cortical-column model. a**, Information detection increases with increasing values of synaptic upscaling ratio, $\beta_{inter}/\beta_{intra}$, during wakefulness when the stimulus intesity is 10 Hz (**a**), 30 Hz (**b**), 70 Hz (**c**) and 90 Hz (**d**). Synaptic upscaling

during wakefulness does not enhance information detection during wakefulness across stimuli compared to those in NREM sleep unless it occurs in DS upscaling. Note that the overall enhancement of information detection as the stimulus intesity increases from **a** to **d**. Error bar corresponds to 95% confidence interval over 10 performance estimate of the K-means clustering algorithms.
(TIF)

**S7 Fig. Dynamical features of spontaneous firing activities in the two-cortical-column model. a**, Spontaneous firing rate signal for a representative trial (i), the distribution of firing rate signals (ii), and the power spectrum of signals (iii) when there is no synaptic upscaling ($\beta_{intra} = 1$, $\beta_{inter} = 1$). **b**, **c**, and **d**, As in **a**, but for when synaptic upscaling is local-selective (LS: $\beta_{intra} = 4$, $\beta_{inter} = 2$), homogeneous (H: $\beta_{intra} = 4$, $\beta_{inter} = 4$), and distance-selective upscaling (DS: $\beta_{intra} = 4$, $\beta_{inter} = 6$), respectively. The dynamical features of spontaneous firing activity in the two-cortical-column model shift from NREM sleep to wakefulness for all synaptic upscaling combinations. Shaded area and Error bar correspond to standard deviation over 500 trials.
(TIF)

**S8 Fig. Effect of intra- and inter-synaptic upscaling on the response of two-cortical-column model to stimuli. a**, Increasing intra-synaptic upscaling while inter-synaptic upscaling is constant (from $\beta_{intra} = 2$, $\beta_{inter} = 2$ to $\beta_{intra} = 6$, $\beta_{inter} = 2$) during wakefulness produces a pulling effect on the amplitude of evoked firing responses in the perturbed (i) and unperturbed cortical column (ii). Conversely, increasing inter-synaptic upscaling while intra-synaptic upscaling is constant (from $\beta_{intra} = 2$, $\beta_{inter} = 2$ to $\beta_{intra} = 2$, $\beta_{inter} = 6$) during wakefulness produces a driving effect on the amplitude of evoked firing responses in the perturbed (i) and unperturbed cortical column (ii). **b**, The amplitude of evoked firing responses increases as the synaptic upscaling transitions from local-selective (LS) to distance-selective (DS) upscaling during wakefulness in the perturbed (i) and unperturbed cortical column (ii). Note that this holds true for other values of stimulus intensity. The amplitude of evoked responses to stimuli in the perturbed (i) and unperturbed cortical column (ii) during wakefulness enhances as synaptic upscaling transition from local-selective (LS) towards distance-selective (DS) upscaling. **c**, Inter-synaptic upscaling increases the net evoked synaptic current, as opposed to when intra-synaptic upscaling increases during wakefulness in the unperturbed cortical column (i). Changes in the net evoked synaptic currrent determine changes in the amplitude of evoked firing responses (ii).
(TIF)

**S9 Fig. Information detection in the two-cortical-column model. a**, Information detection in the perturbed cortical column increases with increasing values of synaptic upscaling ratio, $\beta_{inter}/\beta_{intra}$, during wakefulness when the stimulus intesity is 10 Hz (i), 30 Hz (ii), 70 Hz (iii) and 90 Hz (iiii). **b**, As in **a**, but for the unperturbed cortical column. Error bar corresponds to 95% confidence interval over 10 performance estimate of the K-means clustering algorithms.
(TIF)

**S10 Fig. Robustness of the computational results.** Figures pertain to analysis of evoked firing responses in the perturbed cortical column in the two-cortical-column model. **a**, Information detection for when stimulus intensity is 50Hz (i) and information differentiation (ii) when logistic classification algorithms (see S1 Appendix) are employed. Error bar corresponds to 95% confidence interval over 10 performance estimate of the logistic classification algorithms. Logistic classification algorithms qualitatively replicate the results obtained using K-means clustering algorithms in Fig 4**b**. **b**, Implementing significance tests (see

S1 Appendix) such as student t-test (i) and analysis of variance (ii) qualitatively replicate the results obtain by machine learning techniques pertaining to information detection and information differentiation. **c**, Implementing information theory (see S1 Appendix) manifests that the mutual information between the distribution of evoked responses at stimulus offset and the distribution of stimuli increases as synaptic upscaling transitions from local-selective (LS) to distance-selective (DS) upscaling during wakefulness.
(TIF)

**S1 Appendix. Information quantification.** This conceptual framework was developed to evaluate stimulus-related information in neural signals using machine learning techniques, enabling the systematic extraction and quantification of information encoded in neural activity patterns.
(PDF)

## Acknowledgments

We thank Francesco Damiani for comments on this manuscript.

## Author contributions

**Conceptualization:** Farhad Razi, Belén Sancristóbal.

**Data curation:** Farhad Razi.

**Formal analysis:** Farhad Razi.

**Funding acquisition:** Belén Sancristóbal.

**Investigation:** Farhad Razi, Belén Sancristóbal.

**Methodology:** Farhad Razi, Belén Sancristóbal.

**Project administration:** Belén Sancristóbal.

**Resources:** Farhad Razi, Belén Sancristóbal.

**Software:** Farhad Razi.

**Supervision:** Belén Sancristóbal.

**Validation:** Farhad Razi, Belén Sancristóbal.

**Visualization:** Farhad Razi.

**Writing – original draft:** Farhad Razi.

**Writing – review & editing:** Farhad Razi, Belén Sancristóbal.

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
