## [Decision Letter · Decision Letter 0]

19 May 2025

PCOMPBIOL-D-25-00090

Heterogeneous Synaptic Homeostasis: A Novel Mechanism Boosting Information Propagation in the Cortex

PLOS Computational Biology

Dear Dr. Razi,

Thank you for submitting your manuscript to PLOS Computational Biology. After careful consideration, we feel that it has merit but does not fully meet PLOS Computational Biology's publication criteria as it currently stands. Therefore, we invite you to submit a revised version of the manuscript that addresses the points raised during the review process.

Please submit your revised manuscript within 60 days Jul 19 2025 11:59PM. If you will need more time than this to complete your revisions, please reply to this message or contact the journal office at ploscompbiol@plos.org. Please include the following items when submitting your revised manuscript:

We look forward to receiving your revised manuscript.

Kind regards,

Anna Levina

Academic Editor

PLOS Computational Biology

Lyle Graham

Section Editor

PLOS Computational Biology

**Journal Requirements:**

3) Your manuscript is missing the following section heading: Abstract.  Please ensure that the section heading levels are clearly indicated in the manuscript text, and limit sub-sections to 3 heading levels. An outline of the required sections can be consulted in our submission guidelines here:

5) We notice that your supplementary Figures, and information are included in the manuscript file. Please remove them and upload them with the file type 'Supporting Information'. Please ensure that each Supporting Information file has a legend listed in the manuscript after the references list. Please also cite and label the supplementary figures as  "S1 Figure", S2 Figure" and so forth.

6) Regarding Tables 1 and 2, thank you for stating that they are "adapted from (26). Please include the source details in the tables legends.

7) Thank you for stating in the online submission form that the "Code supporting the findings of this paper are available on github." Please note that your Data Availability Statement is currently missing the DOI/accession number of each dataset OR a direct link to access each dataset. 

8) Please amend your detailed Financial Disclosure statement. This is published with the article. It must therefore be completed in full sentences and contain the exact wording you wish to be published.

2) If any authors received a salary from any of your funders, please state which authors and which funders.

**Reviewers' comments:**

Reviewer's Responses to Questions

Reviewer #1: Comments:

- Cell excitability can differ between wakefulness and the Up state during NREM sleep. The motivation behind calibrating the strength of inhibitory synapses during synaptic upscaling in wakefulness–so that the steady-state average membrane potentials of both pyramidal and inhibitory populations match their respective Up state values during NREM–is not entirely clear. Clarifying this point in the text would be helpful. Furthermore, including a supplementary figure demonstrating that the results are robust to variations in β_GABA would strengthen the analysis.

- Average firing rates during wakefulness are typically comparable to those observed during NREM sleep overall (see works from György Buzsáki), but are substantially lower than the firing rates during UP states within NREM. It would be interesting to know whether the conclusions of the present work would still hold if the model were adjusted to preserve this empirically observed relationship among firing rates across brain states.

- Several figures are difficult to read due to missing legends and unclear parameter annotations. For example, in Figures 2d and 3a, it would be helpful to explicitly indicate the values of β_intra and β_inter within the figure panels. This suggestion extends to other relevant figures as well.

- To enhance self-containment and reproducibility, please consider including the mathematical definitions of the metrics used to quantify information detection and differentiation in the Methods section.

Reviewer #2: In this manuscript the authors present a compelling and novel hypothesis regarding the role of heterogeneous synaptic homeostasis in boosting information propagation in the cortex during wakefulness compared to NREM sleep. The authors propose that the differential upscaling of excitatory connections, with stronger upscaling between different cortical areas (inter-excitatory) compared to within individual areas (intra-excitatory), is crucial for the enhanced transfer of neural responses and information observed during wakefulness. This contrasts with the uniform synaptic upscaling suggested by the synaptic homeostasis hypothesis (SHY).

Using a Wilson-Cowan model of cortical columns, the authors simulate neural dynamics during sleep and wake states by adjusting excitatory coupling and potassium currents. They test how varying intra- and inter-area synaptic scaling (βintra and βinter) affects evoked responses and information transmission, introducing local-selective, homogeneous, and distance-selective configurations. Their results show that distance-selective upscaling (βinter > βintra > 1) most effectively enhances response amplitude and information flow, especially downstream. These findings are supported by statistical and machine learning analyses, and suggest a possible neuromodulatory mechanism underlying wake-sleep differences.

This is a computational study that highlights a potentially interesting mechanism to account for information processing changes in the transition from wakefulness to sleep. However, I cannot recommend it for publication until some concerns are addressed.

1. Clarification of Statement on Stimulus-Evoked Propagation Patterns 

The authors state: "During wakefulness, neural responses to external stimuli exhibit a broader spatiotemporal propagation pattern compared to deep sleep." While this generalization is supported by several studies, it oversimplifies the complexity of stimulus-evoked dynamics across brain states. For example, evidence from [5] (2022, PNAS, https://doi.org/10.1073/pnas.2021252118) demonstrates that somatosensory stimulation during sleep can evoke bilateral activation in primary somatosensory cortices. The authors are encouraged to review and nuance this statement in the introduction to reflect a more accurate and comprehensive view of the literature.

2. Parameter Sensitivity and Generalizability of Results

 The model presented includes numerous parameters, and the outcomes may be highly sensitive to the specific values chosen. Although the authors base their parameter selection on previous literature (Tables 2–5), it remains unclear how robust the findings are to variations in these values.This does not undermine the validity of the proposed mechanism, but it does suggest that different parameter configurations could produce varying outcomes—or that similar outcomes might arise from alternative configurations. Additionally, while the authors refer to the excitatory synapses as being in a "balanced" state, a clearer definition and explanation of how this balance is achieved would strengthen the study’s clarity and reproducibility. The lack of direct experimental validation of the heterogeneous synaptic upscaling mechanism, while acknowledged, further emphasizes the need for caution in generalizing the findings.

3. Definition and Methodological Clarity for NMI

Normalized Mutual Information (NMI) appears in the caption of Figure 3 before it is defined in the text. It is recommended that the term be introduced and explained prior to its use. While the authors commendably share the code for computing information detection and differentiation, these concepts should be thoroughly described in the methods section to ensure transparency and reproducibility.

4. Phase Diagram Enhancements and Reference Inclusion 

The phase diagram currently reports only the NMI values. It would be beneficial to also include information regarding the emergent dynamics of network activity in this space—specifically, the regions where slow oscillations occur. This would align the model more closely with empirical studies and facilitate interpretation. The authors are encouraged to cite and discuss related work, such as [1], which employs a similar analysis approach.

5. Literature Contextualization and Additional References

 The statement: "Most studies in sleep research have not explicitly evaluated information content and are primarily based on the amplitude of evoked neural signals," would benefit from appropriate citations to support this claim. The authors should distinguish between studies that focus on signal amplitude and those that address information content. To further strengthen the manuscript, the following references are recommended for inclusion due to their relevance to sleep-state dynamics and local/global network interactions:

[1] Tort-Colet, N., Capone, C., Sanchez-Vives, M. V., & Mattia, M. (2021). Attractor competition enriches cortical dynamics during awakening from anesthesia. Cell Reports, 35(12).

[2] Andrillon, T., & Oudiette, D. (2023). What is sleep exactly? Global and local modulations of sleep oscillations all around the clock. Neuroscience & Biobehavioral Reviews, 155, 105465.

[3] Miyazaki, T., Kanda, T., Tsujino, N., Ishii, R., Nakatsuka, D., Kizuka, M., ... & Yanagisawa, M. (2020). Dynamics of cortical local connectivity during sleep–wake states and the homeostatic process. Cerebral Cortex, 30(7), 3977-3990.

[4] Marsh, B., Navas-Zuloaga, M. G., Rosen, B. Q., Sokolov, Y., Delanois, J. E., González, O. C., ... & Bazhenov, M. (2024). Emergent effects of synaptic connectivity on the dynamics of global and local slow waves in a large-scale thalamocortical network model of the human brain. PLOS Computational Biology, 20(7), e1012245.

[5] Rosenthal, Z. P., Raut, R. V., Bowen, R. M., Snyder, A. Z., Culver, J. P., Raichle, M. E., & Lee, J. M. (2021). Peripheral sensory stimulation elicits global slow waves by recruiting somatosensory cortex bilaterally. Proceedings of the National Academy of Sciences, 118(8), e2021252118.

Reviewer #3: In the present manuscript, Razi and Sancristóbal analyze a neural mass model that mimics the activity of a cortical column, along with its extension to two interacting cortical columns, incorporating synaptic upscaling to explain response properties across different brain states—specifically, awake-like and NREM sleep-like regimes. Their results show that upscaling excitatory connections not only induces dynamic changes in network activity reminiscent of these two brain states but also alters the efficiency of information propagation across networks. Notably, they report that stimulus information propagates more efficiently during wakefulness than during NREM sleep, provided that synaptic upscaling between distinct networks exceeds that of local and recurrent connections.

I enjoyed reading the article; it is very clear, the methods are well described, and the results are thoroughly justified and presented. I have only minor concerns that, if addressed, could further strengthen the manuscript:

1) A more detailed dynamical systems analysis should be provided, at least for the one-column model. It appears that the awake-like activity is modeled as a stable node, while the NREM sleep-like activity is represented by a limit cycle driven by self-excitation and adaptation. This distinction should be clarified using standard linear stability analysis.

2) It is unclear whether the stimulation in the NREM sleep regime is applied during the upstate, the downstate, or if this distinction is not relevant. Clarification on this point would be helpful.

3) Could the authors comment on why they did not use Fisher information to quantify the discriminability between closely spaced stimuli? This seems like a natural choice for such an analysis.

4) Finally, the authors should discuss their findings in relation to the work by Pachitariu et al. (2015, DOI: 10.1523/JNEUROSCI.3318-14.2015).

I hope the authors find these comments helpful.

**Have the authors made all data and (if applicable) computational code underlying the findings in their manuscript fully available?**

Reviewer #1: Yes

Reviewer #2: Yes

Reviewer #3: Yes

PLOS authors have the option to publish the peer review history of their article (what does this mean?). If published, this will include your full peer review and any attached files.

Reviewer #1: No

Reviewer #2: No

Reviewer #3: No

**Figure resubmission:**
---

## [Decision Letter · Decision Letter 1]

5 Aug 2025

Dear Razi,

We are pleased to inform you that your manuscript 'Heterogeneous Synaptic Homeostasis: A Novel Mechanism Boosting Information Propagation in the Cortex' has been provisionally accepted for publication in PLOS Computational Biology.

Best regards,

Anna Levina

Academic Editor

PLOS Computational Biology

Lyle Graham

Section Editor

PLOS Computational Biology

Reviewer's Responses to Questions

**Comments to the Authors:**

Reviewer #1: The authors have addressed all my concerns raised in the previous round.

Reviewer #2: The authors have made a clear effort to address my comments and concerns, including running additional experiments to support their claims. Their responses were thorough and satisfactory. I have no further questions, and I recommend the paper for publication.

Reviewer #3: The authors have thoroughly addressed all the concerns I raised in my previous review.

**Have the authors made all data and (if applicable) computational code underlying the findings in their manuscript fully available?**

Reviewer #1: None

Reviewer #2: Yes

Reviewer #3: Yes

PLOS authors have the option to publish the peer review history of their article (what does this mean?). If published, this will include your full peer review and any attached files.

Reviewer #1: No

Reviewer #2: No

Reviewer #3: No

---

## [Editor Report · Acceptance letter]

PCOMPBIOL-D-25-00090R1

Heterogeneous Synaptic Homeostasis: A Novel Mechanism Boosting Information Propagation in the Cortex

Dear Dr Razi,

I am pleased to inform you that your manuscript has been formally accepted for publication in PLOS Computational Biology. Your manuscript is now with our production department and you will be notified of the publication date in due course.

With kind regards,

Anita Estes
